# *Chenopodium quinoa*’s Ingredients Improve Control of the Hepatic Lipid Disturbances Derived from a High-Fat Diet

**DOI:** 10.3390/foods12173321

**Published:** 2023-09-04

**Authors:** Aurora Garcia Tejedor, Claudia Monika Haros, José Moisés Laparra Llopis

**Affiliations:** 1Bioactivity and Nutritional Immunology Group (BIOINUT), Faculty of Health Sciences, Universidad Internacional de Valencia—VIU, Pintor Sorolla 21, 46002 Valencia, Spain; agarciate@universidadviu.com; 2Instituto de Agroquímica y Tecnología de Alimentos (IATA), Consejo Superior de Investigaciones Científicas (CSIC), 46980 Valencia, Spain; cmharos@iata.csic.es; 3Molecular Immunonutrition Group, Madrid Institute for Advanced Studies in Food (IMDEA-Food), 28049 Madrid, Spain

**Keywords:** quinoa, macrophage, innate lymphoid cells, high-fat-diet-induced obesity, lipid homeostasis

## Abstract

This study explored the effects of *Chenopodium quinoa*’s ingredients on the major lipids’ hepatic profile and the functional selective differentiation of monocyte-derived macrophages and innate lymphoid cells in mice on a high-fat diet. Six-week-old Rag2^-/-^ and Rag2^-/-^Il2^-/-^ mice received (12 days) a low-molecular-weight protein fraction (LWPF) or the lipid fraction (qLF) obtained from the cold pressing of *C. quinoa*’s germen. At the end of the experiment, mouse serum and liver tissue were collected. The differences in triglycerides, phospholipids, and the major lipids profile were analyzed. Infiltrated monocyte-derived macrophages and innate lymphoid cells (ILCs) and the expression of liver metabolic stress-related mRNA were measured. In the Rag2^-/-^ mice, feeding them LWPF appeared to improve, to a larger extent, their hepatic capacity to utilize fatty acids in comparison to the qLF by preventing the overwhelming of triglycerides (TGs), despite both reducing the hepatic lipid accumulation. An analysis of the hepatic major lipids profile revealed significant increased variations in the PUFAs and phospholipid composition in the Rag2^-/-^ mice fed with the LWPF or LF. The Rag2^-/-^Il2^-/-^ mice, lacking innate and adaptive lymphocytes, seemed resistant to mobilizing hepatic TGs and unresponsive to lipid accumulation when fed with the LF. Notably, only the Rag2^-/-^ mice fed with the LWPF showed an increased proportion of hepatic CD68+F4/80+ cells population, with a better controlled expression of the innate immune ‘Toll-like’ receptor (TLR)-4. These changes were associated with an oriented expansion of pluripotential CD117+ cells towards ILC2s (CD117+KLRG1+). Thus, *C. quinoa*’s ingredients resulted in being advantageous for improving the mechanisms for controlling the hepatic lipotoxicity derived from a high-fat diet, promoting liver macrophage and ILCs expansion to a selective functional differentiation for the control of HFD-driven immune and metabolic disturbances.

## 1. Introduction

Nutrition begins with food, which can differentially impact human health. Nutrients (i.e., glucose, fatty acids, and amino acids) are closely linked to immune cells’ fate and function, influencing rapid and extensive changes in their activities at different stages of life [1]. Foods provide several ingredients with biological functions beyond their mere nutritional role, for example, with the capacity to interact with innate immune receptors, whose activity is closely linked to the direction and severity of inflammatory and immune disorders [2]. Thus, the metabolic reprogramming of the innate monocyte/macrophage population and lymphoid cells’ biology appears as a distinctive stamp in metabolic diseases [3,4,5].

In western societies, non-communicable diseases (NCDs) maintain a continuous increase, affecting an estimated 70% of all deaths worldwide, paralleling the worldwide increases in cardiovascular diseases, cancers, chronic pulmonary diseases, and type 2 diabetes (T2D). In this context, non-alcoholic fatty liver disease (NAFLD) incurs a high risk for the development of T2D and other major features of metabolic syndrome (mainly cardiovascular complications such as atherosclerosis and myocardial ischemia/infarction), taking a toll regarding liver-related morbidity and mortality. NAFLD has become one of the most common liver pathologies worldwide, affecting an estimated 15–30% of most populations [6]. NAFLD comprises a spectrum ranging from simple steatosis to steatohepatitis, which can progress to liver cirrhosis and hepatocellular cancer. Lipid overload is a primary factor triggering hepatic lipotoxicity and injury. In this context, innate immunity has been revealed as critical to determining lipid accumulation and handling the promotion of metabolic homeostasis [7]. While an important link between immune and metabolic signals has been established, the relationship between dietary immunonutritional ingredients and their influence on the functional differentiation of key innate immune effectors is less clear.

*Chenopodium quinoa*’s seeds constitute a nutritious and rich source of carbohydrates, high-quality proteins, fiber, and microelements, but also of immunonutritional serine-type protease inhibitors (SETIs) [8,9]. These compounds, taken at the dietary nutritional recommendation (i.e., 0.4 g/kg body weight), have been shown to be effective in increasing the proportion of intrahepatic macrophages, promoting their selective and functional adaptation [8]. More recent research has identified liver macrophages as determinant players in the diet-regulated control of hepatic energy storage and fat accumulation [5]. In addition, intestinal innate lymphoid cells (ILCs) have been revealed as key contributors to diet-induced obesity [4]. This preliminary research suggested that the acquisition of specific phenotypes by ILCs seems to be responsible for the integration of immune and metabolic signals for modulating hepatic fat accumulation. Lipid utilization could have important effects, worsening or improving inflammatory response(s) and their resolution during the adequate development of immune response(s) [10]. These studies revealed the important effects that foods’ ingredients can exert on the maturation and differentiation, as well as function, of innate immune effectors, which result in being critical for the development of NCDs. Notwithstanding, the relationship between hepatic macrophages and ILCs’ maturation with the administration of SETIs remains incomplete.

In view of the aforementioned, the objective of this study is to explore the immunonutritional effects of different ingredients of *Chenopodium quinoa*, a low-molecular-weight protein fraction (LMWF) enriched with serine-type protease inhibitors (SETIs) and a lipid extract (qLF), to preserve high-fat-diet-induced imbalances in hepatic metabolic homeostasis and innate immunity. 

## 2. Material and Methods

### 2.1. Isolation of Chenopodium Quinoa’s Ingredients

*C. quinoa* seeds were processed to obtain the lipid fraction (qLF) via cold pressing and germs via wet milling [8]. In order to obtain the fatty acid profile, the lipids of the samples were transesterified to convert them into fatty acid methyl esters [11]. The seeds were mixed (5 mL) with a reagent containing methanol:heptane:toluene:2,2-dimethoxypropane:H_2_SO_4_ (39:34:20:5:2, by vol). The samples were heated at 80 °C to be digested and to attain the transesterification in a single step. Isolated fatty acids were analyzed using GC-MS. To obtain germs, the seeds were steeped in a sulfur dioxide solution 0.25% at pH 5.0, adjusted with lactic acid, for 1.25 h at 30 °C. The steeped quinoa was milled using a plate mill to separate the germ fraction via flotation in water. The extraction procedure was performed by cold pressing the quinoa germ in a press with a rod of stainless steel.

To obtain the bioactive LMWF, the germs were finely ground and homogenized prior to being mixed with 5 volumes of phosphate-buffered solution (PBS) (0.01 M phosphate buffer, 0.0027 M potassium chloride, and 0.137 M sodium chloride, pH 7.4). The mixtures were maintained with vigorous vortexing for 1h at room temperature. The samples were centrifuged (6000 rpm/10 min/4 °C) and the supernatants were thermally treated (60 °C/30 min). After a subsequent centrifugation (6000 rpm/10 min/4 °C) of the supernatants, they were filtered (30 kDa MWCO, Amicon^®^, Darmstadt, GER). This fraction resulted in enriched serine-type protease inhibitors (SETIs) (<13 kDa) carrying a glycosidic prosthetic group [9,12]. RP-LC-MS/MS analyses confirmed the absence of bacterial lipopolysaccharide in the extracts of the LWPF using LPS from *E. coli* (L3129, Sigmaaldrich, Darmstadt, GER) as a reference [12].

### 2.2. Experimental Model

The animal experiments were conducted in accordance with the guidelines of the Institutional Animal Care and Use Committee of the Spanish High Research Council of Research (CSIC) and regional government of Madrid (Spain) (Proex 298.0/21). For the studies, both Rag2^-/-^ mice from a C57BL/6 background and Rag2^-/-^IL2^-/-^ mice kindly donated by Dr. Balbino Alarcón (CBM-CSIC) were used. These models were developed and have been characterized elsewhere [13,14]. The animals (5 weeks old) were housed at 22 (22–24 °C) and allowed to adapt to a 42 kcal% high-fat diet (IN93G mod, irradiated, Ssniff spezialdiäten gmbh) one week before starting the study period (15 days). The animals (6-week-old) maintained under a high-fat diet (HFD) received the LWMF (100 µg/20 g) and lipid fraction (0.8 µL/20 g) every 2 days, for 15 days, by gavage. These amounts were calculated based on previous research [9] and the nutritional recommendations for bread consumption. The control animals received 100 µL of a saline solution. Longer periods reduced the analytical capacity to determine the differential effects of feeding the animals *C. quinoa*’s ingredients. 

Body weight gain and food consumption were measured every 2 days. Whole blood was collected in EDTA-treated tubes and centrifuged (6000 g/10 min) to obtain clear supernatants. Different liver sections (30–50 mg) were immersed in RNA later (Qiagen, Hilden, Germany), Krebs’s, or RIPA buffer and kept at −80 °C until analysis. Intra-colon (transversal) fecal samples were collected and immediately frozen for microbial determinations.

### 2.3. Biochemical Analyses

The blood’s clear supernatants were used for glucose (KA1648, Abnova, Taoyuan, Taiwan) and insulin (RAB0817-1KT, Sigma-Aldrich, Darmstadt, GER) quantification, as well as triglycerides (T2449, Merck-Sigma, Darmstadt, GER), according to the manufacturer’s instructions. The Homeostatic Model Assessment of Insulin Resistance (HOMA-IR) value was used to define insulin resistance according to the following formula: (insulin (μIU/mL) × glycaemia (mg/dL))/405.

### 2.4. Immunephenotyping of Hepatic Cells

Liver aliquots kept in Krebs’s buffer were immersed in 0.5 mL of a trypsin/EDTA mixture (T4049, Merck-Sigma). The samples were incubated (37 °C/15 min) with gentle shaking and large particles from the tissue sections were completely disaggregated using pre-separation cell strainers (70 µm). After centrifugation (1200 rpm/5 min), the cells were collected in PBS and mixed with the following fluorochrome-conjugated anti-mouse antibodies, (1) CD11b (Cat. No. BS-1014R, Thermo-Fisher, Madrid, Spain), CD68 (Cat. No. F-2761, Thermo-Fisher), F4/80 (Cat. No. MF48005, Invitrogen, Waltham, MA, USA), and TLR4 (Cat. No. 558294, BD Bioscience, Madrid, Spain), and (2) CD117 (553869, BD Bioscience, MAdrid, Spain), KLRG1 (Cat. No. 564013, BD Bioscience), Nkp46 (Cat. No. 740822, BD Bioscience), and CD56 (Cat. No. 748098, BD Bioscience, Madrid, Spain). Then, the samples were prepared for a flow cytometry analysis with the Immunoprep kit (Beckman Coulter, Pasadena, CA, USA) and further analyzed on a SORP Flow Cytometer (BD Biosciences, Madrid, Spain).

### 2.5. ^1^H-NMR Analysis

Analyses were performed at Centro de Investigaciones Biomédicas ‘Alberto Sols’ (IIB-CSIC, Madrid Spain). HR MAS spectra were acquired from a 11.7 Tesla Bruker Avance spectrometer operating at 500.113 MHz at 4 °C with a 5 KHz spinning rate. 1D 1H HR MAS spectra were acquired with Nutrients 2020, 12, 1946 4 of 15 using the Carr Purcell Meiboom Gill (CPMG) sequence with 2 s of water pre-saturation, 144 ms of echo time, and 128 scans; the data were collected into a 32 K data point using a spectral width of 10 KH (20 ppm) and water pre-saturation during a relaxation delay of 2 s and a 1D sequence for diffusion measurement using a stimulated echo using bipolar gradient pulses for diffusion (stebpgp1s1d), with big delta 200 ms, little delta 1.2 ms, and a sine-shaped gradient followed by a 300 us delay for gradient recovery, a 5 kHz spectral width, a 32 K data point, and 128 scans.

The quantification of the metabolites detectable in the spectra was performed by measuring the area of the peaks using MestReC software (Mestrelab Research, Santiago de Compostela, Spain); the data were manually phased and baseline corrected. NMR spectra were referenced to the FA terminal -CH_3_ signal at δ, 0.89 ppm. The analysis of the metabolites was performed by selecting the following resonances: methyls (-CH_3_) at 0.89 ppm (saturated FA chains), -CH_3_ at 0.96 ppm (n-3 EPA and DHA), acyl chains methylenes (CH_2_) n at 1.33 ppm, CH_2_-C-CO at 1.58 ppm, CH_2_C=C at 2.02 ppm, CH_2_CO other than DHA at 2.25 ppm, CH_2_CO of DHA at 2.33 ppm, =C-CH_2_-C= at 2.78 ppm, and CH=CH at 5.33 ppm.

### 2.6. Microbiological Analyses

The composition of the microbiota was analyzed using real-time PCR [15]. Genus-, group-, and species-specific primers were used as described previously to quantify the different bacterial groups of the intestinal microbiota.

### 2.7. Cell Culture

Human macrophage-like cells (HB-8902^®^) were obtained from the ATCC (USA). For the experiments, the cells were seeded (3 × 10^4^ cells/cm^2^) with 1 mL of Earl’s Modified Eagle Medium (EMEM) onto 12-well plates and returned to the incubator for an additional 24 h. Solutions of the LMWF (100 µg/mL) and qLF (0.8 µL/mL), diluted in EMEM, were exposed to cell cultures along the mito-stress assay (4 h/37 °C).

### 2.8. Cell ‘mito’ Stress Test Assay

Human-like macrophage (HB-8902^®^) cells were obtained from the ATCC and grown according to the recommended conditions. Twenty-four hours before the experiments, the cell cultures were plated in a Seahorse 96-well plate (4 × 10^4^ cells/well) and returned to the incubator. An extracellular flux analyzer (Seahorse Biosciences, North Billiberica, MA, USA) was used to assess the oxygen consumption rate (OCR) according to the manufacturer’s recommendations (Cat. No. 103015-100, Agilent, Santa Clara, CA, USA). The basal respiration was initially measured before an injection of oligomycin (2 µM) to determine the ATP-linked OCR. Then, the maximal respiration was estimated by the addition of FCCP (Carbonyl-cyanide-4, trifluoromethoxy, phenylhydrazone) (1.5 µM). Afterwards, a mixture with rotenone (0.5 µM) and Antimycin A (0.5 µM) was used to determine the mitochondrial-independent OCR to correct all the previously obtained measurements [12].

### 2.9. Statistical Analysis

The data were analyzed using the Statgraphics^©^ Plus (version 5.1, Rockville, MD, USA) software. Comparisons between two groups were performed with two-tailed, unpaired *t*-tests. Statistical differences among the various groups were assessed with the Tukey–Kramer test. The Kaplan–Meier method and log-rank test were used for a survival analysis. Statistical significance was set at *p* values of <0.05.

## 3. Results

**Bioactive fractions.** The fatty acid concentrations found in the quinoa’s lipid fraction (qLF) are shown in Table 1. This fraction contained a major proportion of linoleic acid and oleic acid, while stearic acid was found at the lowest proportion. Notably, the ω-3/ω-6 ratio that could be calculated exceeded the 6:1 relationship. Saturated fatty acids were also found, particularly palmitic acid at a significant proportion.

In addition, *Chenopodium quinoa* grains provide a rich protein fraction, within which, in the albumin fraction (up to 15–20%, *w/w*), glycoproteins with molecular weights of less than 10,000 Da can be found. Their composition based on homomeric subunits and chemical structure carrying a carbohydrate-based prosthetic group appear as determinant of their biological attributes [9,16] (Figure 1).

**Hepatic masses and metabolic changes.** The growth parameters and basic metabolic indicators of the hepatic function of the mice fed with quinoa’s components are shown in Figure 2. In the Rag2^-/-^ mice fed with the LMWF, a negative trend was observed in the BW gain during the study period (Figure 2A), which manifested at a higher extent in the Rag2^-/-^Il2^-/-^ mice (Figure 2B). These contrasting effects in the BW gain could be calculated in both models after the administration of the qLF (Figure 2C). Despite significant differences in the hepatosomatic index between the Rag2^-/-^ and Rag2^-/-^Il2^-/-^ mice, either the LMWF or qLF caused a significant reduction in the values calculated for the L/BW ratio (Figure 2D). The initial differences in the L/BW ratio were attributed to differences in the liver masses, independently of the administration of either the LMWF or qLF (Figure 2E). Only the Rag2^-/-^ mice fed with the LMWF displayed opposite variations in their concentrations of liver (Figure 2F) and peripheral TGs (Figure 2G). However, both models showed reduced glycaemia (Figure 2H). These changes allowed for the calculation of significant variations in the TyG index only for the Rag2^-/-^ mice (Figure 2I).

**Major lipids profile.** The ^1^H-NMR analyses revealed reduced hepatic total lipid contents in the Rag2^-/-^ mice receiving either the LMWF or qLF (Figure 3A), where a lower content was quantified in animals receiving the LMWF. Otherwise, the Rag2^-/-^IL2^-/-^ mice fed with the qLF showed much lower total lipid contents than their counterparts receiving the LMWF. Significant increases were found, with similar patterns, in the variation in the mobile triglyceride-CH_2_ compounds, suggestive of changes in cholesterol-associated metabolites, in both the Rag2^-/-^ and Rag2^-/-^IL2^-/-^ mice receiving either the qLF or LMWF (Figure 3B). The variation in mobile triglyceride-CH_3_ compounds occurred at a lesser extent and was not significant in relation to their respective controls (Figure 3C). These data allowed for the calculation of slightly upward trends in the triglyceride-CH_2_/-CH_3_ ratios for the Rag2^-/-^ fed with the LMWF or qLF, while the Rag2^-/-^IL2^-/-^ mice showed slightly upward trends in their mean values for the -CH_2_/-CH_3_ ratios only in those animals receiving the LMWF (Figure 3D). These variations in the -CH_2_/-CH_3_ signals may constitute metabolic adaptations for better controlling the cholesterol metabolism, which could help to prevent a proinflammatory orientation of innate immune cell effectors. ^1^H-NMR enabled the quantification of significant increases in the MUFA and PUFA contents in the Rag2^-/-^ mice, while only the MUFAs increased in the Rag2^-/-^IL2^-/-^ mice, despite the treatment (Figure 3E,F). These variations allowed for the calculation of significant reductions in the PUFA/MUFA ratios in both models (Figure 3G). Only the Rag2^-/-^ mice displayed increased proportions of compounds carrying the saturated =CH-CH_2_-CH_2_ group, despite the quinoa ingredient they received (Figure 3H). Otherwise, both genotypes showed increased contents of compounds carrying the unsaturated =C=CH-CH= group (Figure 3I).

**Hepatic innate immunity.** To determine the impact of feeding the quinoa ingredients on the phenotype’s adaptations of monocytes and ILCs, the expression of macrophage/monocyte-selective markers and innate lymphoid precursors infiltrated into the livers of the HFD-fed mice was monitored (Figure 4). An analysis of the innate lymphoid precursors evidenced the plasticity and expansion of three subsets representative of pluripotential ILCs (CD117^+^NKp46^-^CD56^-^KLRG1^–^) and precursors of ILC2s (CD117^+^KLRG1^+^) and ILC3s (CD117^+^NKp46^+^CD56^+^) [17]. In the Rag2^-/-^ mice, feeding them the LMWF increased the proportion of all three identified subsets of ILCs (Figure 4A–C). When feeding them the qLF, only significant increases in ILC2s and ILC3s cells were quantified. In the Rag2^-/-^IL2^-/-^ mice, the administration of the LMWF or qLF only appeared to increase the proportion of ILC2s and ILC3s cells. These results support the potential of *C. quinoa*’s ingredients for promoting a differential functional polarization of key innate immune effectors such as the macrophages and ILCs involved in lipid accumulation. 

The administration of the qLF favored higher proportions of intrahepatic monocytes in both the Rag2^-/-^ and Rag2^-/-^IL2^-/-^ mice (Figure 4D), whereas only the Rag2^-/-^IL2^-/-^ mice displayed lower proportions of CD11b^+^F4/80^+^ cells in comparison their respective controls. Otherwise, after feeding them the LMWF, the relative abundances of CD68^-^F4/80^+^ and CD68^+^F4/80^+^ cells were higher only in the Rag2^-/-^ mice (Figure 4E,F). However, contrasting patterns for CD68^-^F4/80^+^ and CD68^+^F4/80^+^ cells were identified in the Rag2^-/-^IL2^-/-^ mice. A significantly increased expression of TLR4 in the Rag2^-/-^ mice fed with the qLF could be identified (Figure 4G). Notably, the administration of either the LMWF or qLF decreased the TLR4 expression in the Rag2^-/-^IL2^-/-^ mice. Further studies were performed in vitro to demonstrate the impact of the LMWF and qLF on the bioenergetic adaptations of human macrophage-like (hMθs) cells (Figure 4H,I). Analyses of the mitochondrial respiration revealed that both fractions tested suppressed the bioenergetics of hMθs, preserving maximal cell respiration (Figure 4K), which altogether supports the acquisition of a M1-like phenotype.

**Microbiota composition.** To examine the potential effects of the LMWF and qLF on changes in the fecal groups of the microbiota, the compositions of those in the Rag2^-/-^ and Rag2^-/-^IL2^-/-^ mice were compared (Figure 5). The administration of both the LMWF and qLF caused a reduction in the proportion of bacterial phyla in the Rag2^-/-^ or Rag2^-/-^IL2^-/-^ mice in relation to their controls (Figure 5A). In the Rag2^-/-^ mice, feeding them the LMWF reduced their proportions of *Firmicutes* (Figure 5B) and *Bacteroidetes* (Figure 5C), while the qLF appeared not to modify these populations. In the Rag2^-/-^IL2^-/-^ mice, contrasting patterns could be observed when feeding them the LMWF or qLF. These bacterial changes were majorly reflected in changes in the *Firmicutes/Bacteroidetes* ratio in the Rag2^-/-^ mice (Figure 5D).

When considering specific groups of bacteria, particularly those considered to be beneficial for human health, *Bifidobacterium* spp were not affected in the Rag2^-/-^ mice (Figure 5E), but *Lactobacillus* spp were reduced (Figure 5F). This bacterial genus was increased in the Rag2^-/-^ mice fed with the LMWF, but slightly reduced in animals receiving the qLF. These bacterial changes could be associated with variations in the family *Enterobacteriaceae* only in the Rag2^-/-^IL2^-/-^ mice (Figure 5G). The data llowed to calculate different variations in the Bifidobacterium spp/Enterobateriaceae groups (Figure 5H) The data could be translated into significant changes in the *Bifidobacterium spp/Enterobacteriaceae* ratio, which appeared to be a sensitive marker for the administration of the LMWF in both mice genotypes.

## 4. Discussion

This study showed the immune and metabolic potential of the isolated ingredients of *C. quinoa* for modulating the hepatic innate immunity in experimental preclinical models, which allows for the identification of the integrative roles of intrahepatic ILCs and monocyte-derived macrophages in the prevention of homeostatic alterations caused by a high-fat diet. The study expands the knowledge on the role of innate biology by providing evidence that defined *C. quinoa*’s LMWF and qLF as modulating human macrophage-like cells’ bioenergetics, acquiring a similar phenotype to that identified in in vivo assays. Therefore, the administration of *C. quinoa*’s LMWF and qLF can represent promising immunonutritional strategies for preventing chronic-metabolic-disease-associated comorbidities. Metabolic reprogramming is a hallmark of chronic metabolic diseases, in which nutritional strategies might play a key role. These effects are aligned with the profound changes that nutrition undergoes; from its classical perspective, this ensures the supply of nutrients towards precision nutrition with a greater consideration of nutrients, in order to influence the dynamics of homeostatic balances.

Previous research has shown that the inclusion of *C. quinoa*’s flour in bread formulations (up to 20%, *w/w*), partially replacing wheat, is effective in controlling and reducing the potential high-fat-diet-induced alterations in the hepatosomatic index (Liver to BW ratio), insulin resistance (HOMAir), and peripheral TGs levels in C57Bl/6 mice [16]. These effects could be associated with positive variations in the peripheral proportion of the innate immune myeloid population. Previous research [16] has demonstrated that the inclusion of *C. quinoa*’s flour in bread formulations allows for the modulation of the myeloid population. However, this effect occurred at a much lower extent to that found in this study, which clarifies that these effects can be attributed mostly to the LMWF rather than to any other ingredient.

Furthermore, while previous studies have contributed to identifying a link between *C. quinoa*’s ingredients and hepatic macrophage polarization and lipid homeostatic signals, the participation of ILCs in these processes remains unsolved. Here, feeding mice the LMWF promoted a selective expansion of monocyte CD68^+^F4/80^+^ cells, as well as specific lymphoid subsets such as ILC2s and ILC3s. The macrophage differentiation towards a CD68^+^F4/80^+^ phenotype can be positively associated with reduced hepatic lipid accumulation [5], while the ILCs’ subsets appeared to mediate the integration of immunonutritional signals into hepatic lipid homeostasis. The administration of the LMWF to the animals appeared to be necessary to develop a hepatic selective expansion of the myeloid population, with a more controlled expression of F4/80^+^TLR4^+^ biomarkers (Figure 3F,G). This could have important consequences preventing the excessive activation of innate immune signaling or the aberrant processes contributing to non-resolving inflammation and hepatic fat accumulation. 

Rag2^-/-^ mice display peripheral innate and adaptive immune compositions (5% myeloid cells and 30% lymphoid cells), which better resemble those in humans in comparison to C57Bl/6 mice (3% myeloid cells and 75% lymphoid cells). Thus, the animal model used approaches an intermediary phenotype, where innate immunity acquired more responsibilities as a gatekeeper of the immune response(s) to food ingredients. However, the genetic features of Rag2^-/-^ mice do not appear to be responsible for the lack of a functional induction of the myeloid population carrying the CD68 (mRNA) marker, as this occurs in C57Bl/6 mice fed with bread formulations containing *C. quinoa*’s flour [16]. This assumption is based on the previously reported effects, derived from the administration of a protein extract from *C. quinoa*’s flour, increasing the hepatic proportion of CD68^+^CX3CR1^+^CD74^+^ cells in C57BL/6 mice [18]. Collectively, the data appear to indicate that innate immune signals that stem from the intestinal level result in being significantly important for the hepatic phenotype acquired by infiltrated monocyte-derived macrophages. Hepatic immunity can underlie the link between lipid homeostasis and the risk of metabolic imbalances caused by a high-fat diet [4,5]. From an immunonutritional point of view, ω-6/ω-3 PUFAs exert negative effects on interferon (IFN)-γ signaling due to their capacity to interact and influence the interactions with IFN-γ’s receptor [18]. Thus, the expansion of CD68^+^F4/80^+^ cells cannot be attributed to the fatty acid composition of the qLF administered to the Rag2^-/-^ and Rag2^-/-^IL2^-/-^ mice. The expression of CD68, a marker of inflammation associated with monocytes/macrophages but not dendritic cells [19], is closely associated with IFN-γ signaling [20]. Additionally, PUFA-derived metabolites such as resolvins (i.e., RvD1/D2) display potent anti-inflammatory effects, promoting macrophage polarization towards an M2 end [21]. However, these adaptations could not be recapitulated either in vivo or in vitro. In fact, in vitro studies for monitoring the bioenergetic changes in human-like macrophages challenged with the LP from *C. quinoa* confirmed the M1-like metabolic profile acquired by cells in relation to that of the LMWF (Figure 3H,I).

ILCs have been identified as critical determinants in diet-induced obesity [4]. Particularly, a major role has been attributed to intestinal ILC2s and, to a much lesser extent, to ILC3s, while ILC1s seem to play a more relevant role in adipose tissue. ILC1s participate in the induction of inflammation and development of insulin resistance [22]. Here, the significant reduction in hepatic total lipids in the animals fed with the LMWF is unlikely to suggest the expansion of pluripotential CD117^+^ cells towards a cytotoxic/proinflammatory ILC1s-like phenotype. In fact, preclinical studies have reported protective effects of the expansion of ILC1s in the liver against NAFLD development and its consequences [23]. However, the role of ILC2s still remains imprecise and appears to affect, by either worsening or improving, NAFLD/obesity [24]. Many environmental factors have been studied and most of them appear to be related to ILC2s activation and the production of the profibrotic cytokine IL-13 affecting macrophage selective polarization. ILC2s activation can occur through the uptake of external lipids favoring their proliferation [25]. In this sense, only the Rag2^-/-^ mice fed with the LMWF showed a significantly decreased hepatic TGs fraction (Figure 2F) and total lipids content (Figure 3A) associated with a significant expanded ILC2s proportion (Figure 4F). Furthermore, the increased CD117^+^Nkp46^+^CD56^+^ cell population supports the accumulation and development of an M1-like (CD68^+^F4/80^+^) phenotype by hepatic macrophages [26]. Taken together, these results point to a close functional relationship between macrophages and ILC2s. While macrophages are reported to be responsible for hepatic fat accumulation, ILCs are reported to be critical regulators of tissue macrophage activity and phenotype. Notwithstanding, it appears important to consider the different roles and activities that ILCs can play at the different disease stages of NAFLD/obesity development.

Therefore, the results indicate that the LMWF caused innate immune activation, which, considering its poor susceptibility to the gastrointestinal enzymes, as well as its negligible absorption, could be assumed to stem from the intestinal level and be linked to TLR4 [8,11]. However, these effects were statistically significant in comparison to the control groups and were not translated into pathologic, inflammatory, or cytotoxic signs. In a pathologic context, such as under procarcinogen conditions, it has been reported that feeding with the LMWF can induce cytokine production that potentially contributes to regulating the proinflammatory response(s) [27]. For example, the LMWF could interact with TLR4 signaling via adaptor TRIF/TICAM molecules by skewing the connection of its proinflammatory pathway with the FASN, while modulating macrophage phenotypes and bioenergetics, promoting the production of molecular species such as γ-keto/hydroxy- phosphatidylcholine. Therefore, it is possible that the LMWF used in this study only caused a transient/delayed activation of TLR4, without enhancing the final production of inflammatory mediators derived from FASN activity that lead to chronic inflammatory processes. In addition, the data evidenced significant differences between the immunomodulatory properties of the LMWF and qLF, since the latter appeared to exert inhibitory effects on TLR4 downstream signaling [11,28]. This inhibitory effect was concordant with the changes in the macrophage bioenergetics that were induced by the LMWF and qLF (Figure 4H–J). Macrophage activation undergoes differential orientations depending on cell metabolism, which are associated with the phenotypic and functional changes in macrophages. Proinflammatory macrophages display an increased glycolytic metabolism, while fatty acid oxidation is downregulated. The data evidenced the downregulatory effect that the LMWF and qLF exerted on ATP production. In addition, these changes were accompanied by differential effects on the non-mitochondrial respiration (approach to fatty acid oxidation). Notably, the expression of markers such as F4/80+TLR4+ was associated with M1-like macrophages.

Consistent with previous research, *C. quinoa*’s protein extract exhibited a selective functional macrophage phenotype and ILCs expansion, favoring the HFD-fed Rag2^-/-^ mice exhibiting lower hepatic TGs and an increased CD68^+^F4/80^+^ cells population. Notably, despite the positive effects of PUFAs on immunity and the development of NAFLD risk factors such as obesity, the animals fed with the qLF did not replicate the innate immune response(s) of the LMWF. This study evidenced that feeding with the LMWF promoted higher hepatic concentrations of PUFAs than those with the qLF (Figure 3F–H). A direct relationship between MUFA/PUFA consumption and their hepatic levels was not observed. From a molecular perspective, it could be hypothesized that the LMWF, due to its capacity to interact with TLR4 [11], seems to influence, among others, either the release or activity of the soluble epoxide hydrolase in peripheral blood [29]. This effect could explain, at least in part, the production of important lipid mediators by the macrophage population, with beneficial effects on the hepatic metabolic stress derived from the HFD consumption [29,30]. In the models used, the changes in hepatic PUFAs were associated with the increased CD68^+^F4/80^+^ population only in the Rag2^-/-^ mice carrying ILCs, which supports coordinated regulatory response(s) between both the myeloid and lymphoid innate immune populations. Hepatic immune exhaustion could underlie the link between HFD-induced metabolic stress [31] and losses of liver function at the late stages of NAFLD development, where natural products can exert important effects [32].

The gut microbiota, among others, influence energy expenditure and absorption, fat storage, and low-grade chronic inflammation. The oral administration of the LMWF, in comparison to the qLF, caused common negative variations in the gene copy numbers of the different bacterial groups that were quantified at a higher extent in the Rag2^-/-^ mice (Figure 5). These differences were not associated with high-fat diet consumption and, therefore, could have been due to the induced changes in intestinal innate immunity and its underlying signaling. Notably, the data obtained from genetically modified obese mice on a HFD (8 weeks), supplemented with quinoa (cherry vanilla variety), did not show significant variations in the total counts of Firmicutes and Bacteroidetes [33]. Otherwise, feeding with the protein fraction from *C. quinoa* for a longer period [32] did not seem to promote significant differences in relation to the variations observed in the Rag2^-/-^ mice. However, the obtention of a similar protein fraction from other vegetal sources, such as *Salvia hispanica*, displayed similar effects, but a differential immunonutritional potential from that of *C. quinoa* [32]. These effects were reflected in the significant variations in the gut microbiota Interestingly, in the Rag2^-/-^IL2^-/-^ mice, contrasting patterns were observed, where the qLF appeared to affect the bacterial groups to a higher extent than the LMWF. These alterations were not associated with the increased gut microbiota diversity derived from the high-fat diet consumption [34], where bifidobacterial significantly contribute to the pathophysiological regulation of endotoxemia [35]. The HFD reduced the abundances of *Lactobacillus* spp. and *Bifidobacterium* spp. that appeared to be preserved in the Rag2^-/-^ mice fed with both the LMWF and qLF, while they were increased in the Rag2^-/-^IL2^-/-^ mice fed with the LMWF. Here, the results recapitulate previously reported data demonstrating that the gut microbiota composition appears to be determined by innate immunity [7,36]. The development of selective phenotypes in hepatic myeloid and lymphoid cells cannot be associated with intestinal microbial changes and are thus attributed to the impact of *C. quinoa*’s ingredients on innate immunity (i.e., Toll-like receptors). 

This study demonstrated that *C. quinoa* provides immunonutritional ingredients, such as LMWF and qLF, that promote hepatic innate immunity via the regulation of the major lipids profile. These effects exert beneficial actions in terms of counteracting the immune and metabolic imbalances and liver dysfunction derived from HFD consumption. In this respect, *C. quinoa*’s LMWF and qLF showed advantageous properties, enabling the selective functional differentiation of the hepatic monocyte/macrophage population towards an M1-like phenotype. Additionally, these ingredients influenced the expansion of the precursors of ILCs towards phenotypes with potential benefits for the control of HFD-induced obesity. Thus, the Rag2^-/-^, but not Rag2^-/-^IL2^-/-^, mice developed immune and metabolic protection against hepatic fat accumulation in animals on a high-fat diet. New data are provided regarding the roles played by ILCs and macrophages, as well as their coordination, in the maintenance and integration of innate immune signals for metabolic control. 

## Figures and Tables

**Figure 1 foods-12-03321-f001:**
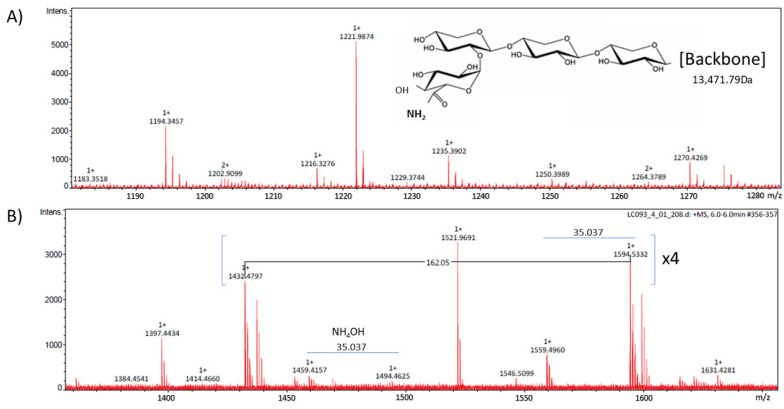
RP-HPLC-ESI-MS/MS analyses (‘m/z’ signals) of the LMWF. (**A**) Identification of the molecular weight corresponding to the protein backbone, and (**B**) identification of mass losses corresponding to the carbohydrate-based prosthetic group [9,16].

**Figure 2 foods-12-03321-f002:**
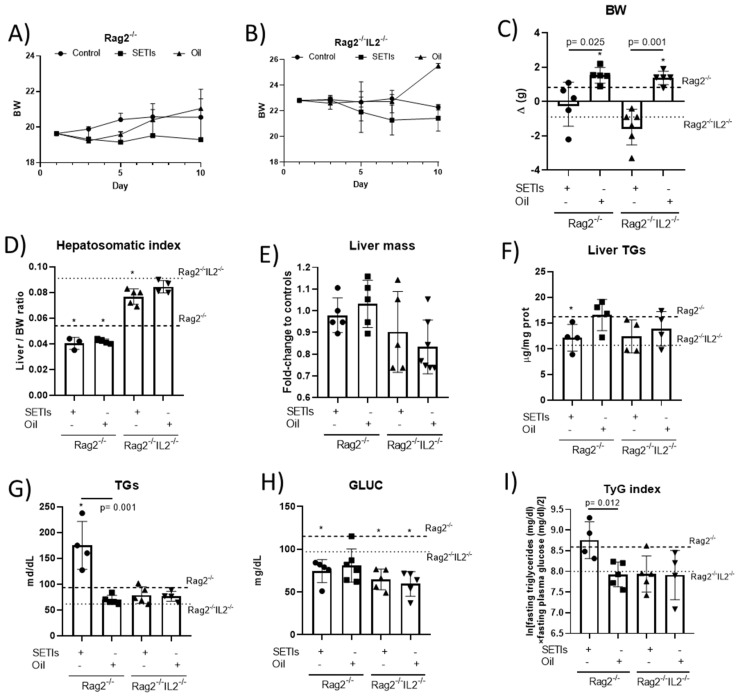
Body weight measures and biochemical parameters in mice administered with the LMWF (SETIs) or the qLF (Oil). (**A**–**D**) Body weight gain, hepatosomatic index (**D**), and liver mass (**E**), liver triglycerides (**F**), serum triglycerides (**G**), glycaemia (**H**) and the Tyg index (**I**) in high-fat diet (HFD)-fed Rag2^-/-^ and Rag2^-/-^IL2^-/-^ mice. Values for control animals under a high-fat diet are represented by dotted lines. Data are presented as mean ± SD (*n* = 4–6). * *p* < 0.05.

**Figure 3 foods-12-03321-f003:**
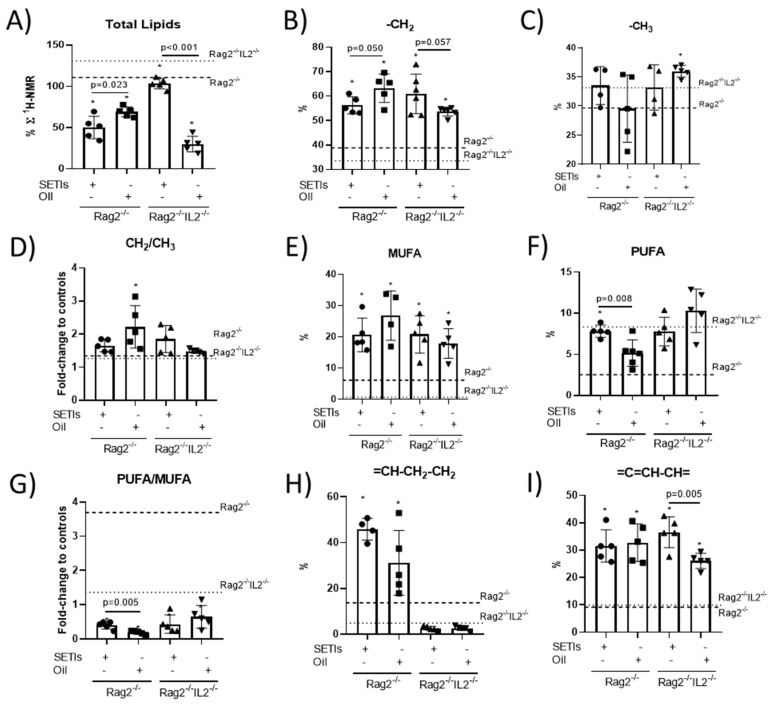
Hepatic major lipids profile in mice administered with the LMWF (SETIs) or the qLF (Oil). ^1^H-MAS MR levels of total lipids (**A**), mobile triglycerides-CH_2_ (**B**) and triglyceride-CH_3_ (**C**) and the CH_2_/Ch_3_ ratio (**D**). (**E**–**I**) Lipid patterns for PUFAs and MUFAs. Values for control animals under a high-fat diet are represented by dotted lines. Data are presented as means ± SD (*n* = 4–6). * *p* < 0.05.

**Figure 4 foods-12-03321-f004:**
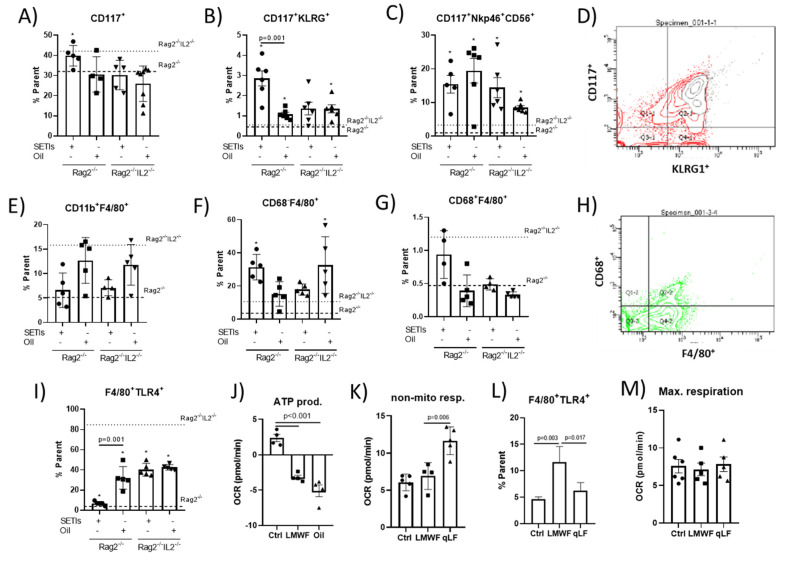
Hepatic immunity in mice administered with the LMWF (SETIs) or the qLF (Oil). Immunephenotyping of hepatic infiltrated innate lymphoid (**A**–**C**), (**D**) typical FACS plot for the identification of ILCs. Immunephenotyping of hepatic infiltrated myeloid (**E**–**G**,**I**), (**H**) typical FACS plot for the identification of myeloid cells. Bioenergetic adaptations in human macrophage-like cells: (**J**), ATP production; (**K**), non-mitochondrial respiration; (**L**), immunephenotyping of HB8902 cells; and (**M**) maximal respiration. Values for control animals under a high-fat diet are represented by dotted lines. Data are presented as means ± SD (*n* = 4–6). * *p* < 0.05.

**Figure 5 foods-12-03321-f005:**
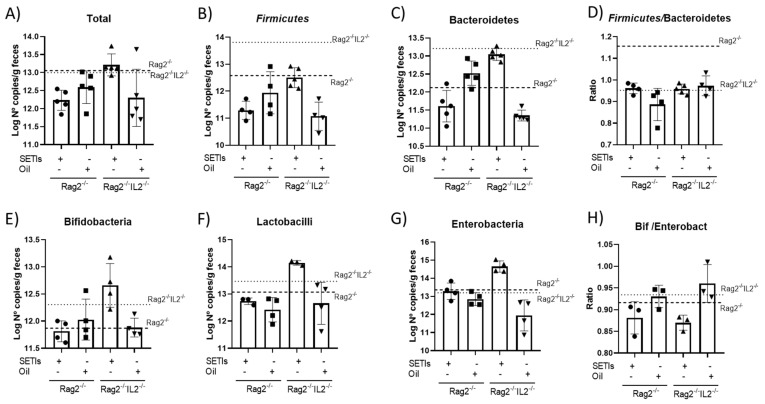
Gut microbiota in mice administered with the LMWF (SETIs) or the qLF (Oil). Microbiota composition of colon content determined by real-time PCR and expressed as log copy number of 16S rDNA gene per gram of feces. Colonic account of different bacterial phyla (**A**–**H**) in mice administered with the LMWF and qLF. Values for control animals under a high-fat diet are represented by dotted lines. Results are expressed as mean ± SD (*n* = 4–6).

**Table 1 foods-12-03321-t001:** Fatty acid composition of quinoa’s lipid fraction.

Fatty Acid	%
Stearic (C18:0)	0.68 ± 0.05
Oleic (C18:1)	32.8 ± 0.3
Linoleic (C18:2)	47.7 ± 1.5
Linolenic (C18:3)	7.12 ± 0.11
Palmitic (C16:0)	11.8 ± 1.4

## Data Availability

The data used to support the findings of this study can be made available by the corresponding author upon request.

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
