# Peer review of "Chenopodium quinoa’s Ingredients Improve Control of the Hepatic Lipid Disturbances Derived from a High-Fat Diet"

_foods, 2023, doi:10.3390/foods12173321_

Round 1

Reviewer 1 Report

After going through the manuscript, i have following queries, i have found it a nice attempt. However, there are few questions that need to be answered from authors 

1.      What is the prevalence of non-communicable diseases (NCDs) in western societies?

2.      At what dietary nutritional recommendation have immunonutritional serine-type protease inhibitors (SETIs) been shown to increase the proportion of intrahepatic macrophages?

3.      How do liver macrophages and intestinal innate lymphoid cells (ILCs) contribute to diet-induced obesity and hepatic energy storage?

4.      What are the potential effects of lipid utilization on inflammatory responses and their resolution during immune response development?

5.      How do the ingredients in Chenopodium quinoa affect the maturation, differentiation, and function of innate immune effectors relevant to the development of NCDs?

6.      In the context of the study's objective, what are the specific immunonutritional effects of different Chenopodium quinoa ingredients, such as the low molecular weight protein fraction (LMWF) enriched with SETIs and the lipid extract (qLF)?

7.      What gaps in knowledge regarding the relationship between hepatic macrophages and ILCs maturation with the administration of SETIs are highlighted in the paragraph?

8.      What processing method was used to obtain the germs from C. quinoa seeds, and how was the lipid fraction (qLF) obtained?

9.      Describe the steps involved in the preparation of the germs for the experiment.

10.  What composition was used for the phosphate buffered solution (PBS) in which the finely grinded germs were mixed?

11.  How long were the mixtures of germs and PBS kept with vigorous vortexing, and at what temperature?

12.  What were the parameters used for centrifugation during the sample preparation?

13.  What thermal treatment was applied to the supernatants after centrifugation, and at what temperature and duration?

14.  Which reference was used for the confirmation of the absence of bacterial lipopolysaccharide in the LWPF extracts?

15.  Why was the study period set at 15 days, and why were longer periods considered impractical?

16.  How frequently were the LWMF and lipid fraction administered to the animals during the 15-day study period?

17.  At what age were the animals used in the experiments, and how long were they allowed to adapt to the high-fat diet before the study period began?

18.  How were comparisons between two groups conducted, and what type of t-test was used?

19.  Why was the Tukey-Kramer's test applied for comparing various groups?

20.  Were the differences observed in the gut microbiota of Rag2-/- mice associated with high-fat diet consumption or induced changes in intestinal innate immunity?

21.  How do the changes in hepatic PUFA levels relate to the increased CD68+F4/80+ macrophage population in Rag2-/- mice carrying ILCs?

22.  How does this study contribute to the understanding of C. quinoa's potential health benefits, particularly in terms of immunonutritional effects and metabolic regulation?

23.  What is the impact of C. quinoa's ingredients on the expansion of precursor cells of innate lymphoid cells (ILCs), and how do these cells contribute to the control of high-fat diet-induced obesity?

Quality of english is good 

Author Response

Reviewer #1

After going through the manuscript, i have following queries, i have found it a nice attempt. However, there are few questions that need to be answered from authors

  1. What is the prevalence of non-communicable diseases (NCDs) in western societies?

According to the World Health Organization (WHO, https://www.who.int/news-room/fact-sheets/detail/noncommunicable-diseases) Noncommunicable diseases (NCDs), also known as chronic diseases, tend to be of long duration and are the result of a combination of genetic, physiological, environmental and behavioural factors. The main types of NCD are cardiovascular diseases (such as heart attacks and stroke), cancers, chronic respiratory diseases (such as chronic obstructive pulmonary disease and asthma) and diabetes.

In the EU, approximately 60% of adults and 20% of school-age children are overweight or obese. Over 1.6 billion people (aged 15 years and above) worldwide are currently either overweight or obese, and this number is predicted to increase to 2.3 billion by 2050 (WHO). Besides, by 2035, it is estimated that there will be >590 million T2D people diagnosed (Diabetes Metab Syndr Obes. 2021; 14: 3567–3602).

Cardiovascular diseases account for most NCD deaths, or 17.9 million people annually, followed by cancers (9.3 million), chronic respiratory diseases (4.1 million), and diabetes (2.0 million including kidney disease deaths caused by diabetes).

  1. At what dietary nutritional recommendation have immunonutritional serine-type protease inhibitors (SETIs) been shown to increase the proportion of intrahepatic macrophages?

As indicated in the manuscript (page 2, line 56), ‘…These compounds at dietary nutritional recommendation (i.e., 0.4 g/kg body weight), have been shown effective to increase the proportion of intrahepatic macrophages promoting their selective and functional adaptation [8]’.

Recommendations have been calculated based on the yield obtained from flours (wheat, C. quinoa, S. hispanica, A. sativa), considering that bread is the main source of flours and it is recommended a dietary intake of 150 g/70 kg.

  1. How do liver macrophages and intestinal innate lymphoid cells (ILCs) contribute to diet-induced obesity and hepatic energy storage?

According to previously published data (Cell Rep. 2019, 28, 202-217; Science 2021, 373.), ILCs determine dietary fat absorption and macrophages exert regulatory roles on liver metabolism as well as lipid clearance, which together influence the hepatic fat accumulation.

  1. What are the potential effects of lipid utilization on inflammatory responses and their resolution during immune response development?

Many thanks for your comment. Immunity and metabolism are interlinked. Macrophages are at the centre of the innate immune response, and the downstream events involving lipid metabolism and inflammation have been mostly studied in these cells upon TLR4 stimulation. Important effects downstream of TLR4 signalling is the regulation of the SREBP-2/LXRaxis, with LXR being considered as the key regulator of the cross-talk between lipid sensing and immune function (Trends Immunol. 2016, 37,819–830). Thus, an adequate communication/regulation of these aspects can significantly contribute to control tissue inflammation and physiological metabolism.

  1. How do the ingredients in Chenopodium quinoa affect the maturation, differentiation, and function of innate immune effectors relevant to the development of NCDs?

Our major hypothesis is that C. quinoa’s ingredients affect intestinal TLR4 signalling and thereby, activate hepatic innate immunity (Bouzas et al., Biomedicines 2021, 9) (page 12, line 381-384). ‘…For example, the LMWF could interact with TLR4 signaling via adaptor TRIF/TICAM molecules by skewing the connection of its proinflammatory pathway with the FASN, while modulating macrophage phenotype and bioenergetics promoting the production of molecular species as g-keto/hydroxy- phosphatidylcholine.’ (page 12, 388-391).

Previous of our own studies have tried to define the underlying molecular signaling in macrophages that results activated when exposed to C. quinoa’s proteins (Cells 2020, 9, 593). These studies identified the TRIF/TICAM pathways as the most probable signaling responsible for the immune/metabolic definition of the macrophage’s function.

  1. In the context of the study's objective, what are the specific immunonutritional effects of different Chenopodium quinoa ingredients, such as the low molecular weight protein fraction (LMWF) enriched with SETIs and the lipid extract (qLF)?

As indicated in previous comments, the most relevant effects would be the control of proinflammatory features of macrophages and thereby helping to control/influence the hepatic metabolism and function.

  1. What gaps in knowledge regarding the relationship between hepatic macrophages and ILCs maturation with the administration of SETIs are highlighted in the paragraph?

From our perspective the gaps revealed are those connecting the maintenance of lipid homeostasis by ILCs to the macrophage phenotype and activity. Previous studies evaluated the role of ILCs (Cell Rep. 2019, 28, 202-217) or macrophages (Science 2021, 373) as individual effectors. Here, it is shown a close link between the presence of ILCs to the maturation of an specific macrophage phenotype. Unfortunately, this study cannot define the molecular signaling of this relationship, which we consider really important.

  1. What processing method was used to obtain the germs from C. quinoa seeds, and how was the lipid fraction (qLF) obtained?

Many thanks for your comment. In view of the reviewer’s comment, a brief description of the isolation and analysis of the lipid fraction (page 2, line 86-90) as well as the method developed by Ballester et al. (Food Hydrocolloids 89:837-843 (2019)) to obtain the germen has been included in the material and methods section (page 2, line 90-94).

  1. Describe the steps involved in the preparation of the germs for the experiment.

We feel sorry for this misunderstanding. In view of the reviewer’s comment, it has been indicated in the text the paragraph corresponding to the preparation of the germs to obtain the LMWF (page 3, line 95).

  1. What composition was used for the phosphate buffered solution (PBS) in which the finely grinded germs were mixed?

We feel sorry for this misunderstanding. As stated in the text (page 2, line 87-88) it was used ‘…phosphate buffered solution (PBS) [0.01 M phosphate buffer, 0.0027 M potassium chloride and 0.137 M sodium chloride, pH 7.4].’

  1. How long were the mixtures of germs and PBS kept with vigorous vortexing, and at what temperature?

We feel sorry for this misunderstanding. As stated in the text (page 2, line 88-89) ‘…Mixtures were kept with a vigorous vortexing for 1h at room temperature.’

  1. What were the parameters used for centrifugation during the sample preparation?

We feel sorry for this misunderstanding. As stated in the text (page 2, line 90-92) ‘…After subsequent centrifugation (6,000 rpm/10min/4ºC) of the supernatants they were fil-tered (30kDa MWCO, Amicon®).’

  1. What thermal treatment was applied to the supernatants after centrifugation, and at what temperature and duration?

We feel sorry for this misunderstanding. As stated in the text (page 2, line 90) ‘…the supernatants thermally treated (60ºC/30min).’

  1. Which reference was used for the confirmation of the absence of bacterial lipopolysaccharide in the LWPF extracts?

We feel sorry for this misunderstanding. As stated in the text (page 2, line 95) it was used ‘…LPS from E. coli (L3129, Sigmaaldrich) as reference’.

  1. Why was the study period set at 15 days, and why were longer periods considered impractical?

The reason to set the study period at 15 days is because in longer periods dimer response(s) are observed for ILCs. Longer study periods revealed similar adaptations in intrahepatic macrophage phenotypes (Bouzas et al., Biomedicines 2021, 9). Our perspective is to use C. quinoa’s ingredients to prevent early changes in innate biology conditioning the risk of suffering immune and metabolic imbalances derived from a high fat diet. Thus, we wanted to evaluate the changes occurring in ILCs and macrophage adaptations at early stages of disease development when animals were put under a high fat diet. In longer study periods we lost the capacity to detect variations in ILCs phenotypes.

  1. How frequently were the LWMF and lipid fraction administered to the animals during the 15-day study period?

We feel sorry for this misunderstanding. As stated in the text (page 3, line 105), C. quinoa’s ingredients were administered ‘…every two days, for 15 days, by gavage’. This dosing pattern was conditioned by the ethics committee.

17.At what age were the animals used in the experiments, and how long were they allowed to adapt to the high-fat diet before the study period began?

We feel sorry for this misunderstanding. As stated in the text (page 3, line 101-103) ‘Animals (5-weeks-old) were housed at 22 (22-24ºC) and allowed to adapt to a 42 kcal% high fat diet (IN93G mod, irradiated, Ssniff spezialdiäten gmbh) one week before starting the study period (15 days).’

18.How were comparisons between two groups conducted, and what type of t-test was used?

The statistical comparisons were conducted for each model considering the administration of the LMWF or qLF. Otherwise, the comparison between models is quite difficult as they do not display the exact genomic background (a limitation of these models). It was used parametric, non-paired, t tests.

  1. Why was the Tukey-Kramer's test applied for comparing various groups?

This test was used because our sample size is not so large and the Tukey method is conservative when there are unequal sample sizes.

  1. Were the differences observed in the gut microbiota of Rag2-/- mice associated with high-fat diet consumption or induced changes in intestinal innate immunity?

Previous studies have shown the predominant role of immunity shaping the gut microbiota and lipid metabolism (Nature 2018, 554(7691):255-259). Unfortunately, this study did not used food ingredients and was difficult to extrapolate the results to human nutrition. Our study tries to fill up this gap of knowledge. Besides, data reveal that adaptation in hepatic immunity does not appear to be associated to changes in the proportion of microbial groups considered beneficial (Bifidobacterium spp and Lactobacillus spp) (page 13, line 446-449). Also, it is shown that lacking ILCs influence the proportion of these genus. Thus, it is concluded that the potential role of gut microbiota is underlined to the administration of immunonutritional food ingredients. Another scenario would be the maintenance of the homeostatic conditions.

  1. How do the changes in hepatic PUFA levels relate to the increased CD68+F4/80+ macrophage population in Rag2-/- mice carrying ILCs?

Many thanks for your comment, which we consider interesting. This link represents a major interest in our research. By the time being, we only can offer a potential ‘theoretical’ link since we still work on that connection. Intestinal macrophages are in close interaction with enteric microbiota (Int. J. Mol. Sci. 2020, 21, 6866), where it has been identified a novel microbiota-dependent crosstalk between macrophages and RORγt (Nat Rev Gastroenterol Hepatol 2019, 16(9):531-543). The latter plays a fundamental role in the regulation of intestinal homeostasis as well as the development of lymphoid tissues such as lymphoid tissue-inducer (LTi) cells and IL-22-producing NKp46+ cells (Nat Immunol 2011, 12(4):320-6). Gut microbiota has been shown to influence the desaturation of hepatic PUFA (Kindt, et al. The gut microbiota promotes hepatic fatty acid desaturation and elongation in mice. Nat Commun 2018;9:3760. doi:10.1038/s41467-018-05767-4). The fact that only Rag2-/- mice display increased PUFA when administered with C. quinoa’s ingredients transpires in the sense that innate immunity, at a higher extent than gut microbiota, influences the level of those. In this context, the functional differentiation of macrophages seems to be critical. Unfortunately, this study cannot demonstrate the relationship between C. quinoa’s ingredients and PUFA levels, which would need a different experimental design.

  1. How does this study contribute to the understanding of C. quinoa's potential health benefits, particularly in terms of immunonutritional effects and metabolic regulation?

This study contributes to establish the close relationship between innate, either lymphoid and myeloid cells, influencing hepatic immunity. Data demonstrate the positive effect of C. quinoa’s ingredients, mainly the LMWF, to influence the differentiation of hepatic macrophages favoring a better control of TGs accumulation in liver tissues when animals are fed a high fat diet.

  1. What is the impact of C. quinoa's ingredients on the expansion of precursor cells of innate lymphoid cells (ILCs), and how do these cells contribute to the control of high-fat diet-induced obesity?

We really appreciate your comment. As stated in a previous comment, the RORgt signaling at intestinal level plays key roles in the maturation of ILCs, and those enable macrophage activation to control the hepatic fat accumulation. We consider important to show such current data to stimulate the multidisciplinary research in the field.

Reviewer 2 Report

The paper entitled ‘Chenopodium quinoa’s ingredients improve control on the hepatic lipid disturbances derived from a high fat diet’ reported the effects of Chenopodium quinoa’s ingredients on major lipids hepatic profile and functional selective differentiation of monocyte-derived macrophages and innate lymphoid cells in mice under a high fat diet. This work in interesting. I have no specific concern but it is mandatory to correct the manuscript in this point:

Please unify whether the Latin name is italicized. C. quinoa in the whole text should be italicized

Author Response

The paper entitled ‘Chenopodium quinoa’s ingredients improve control on the hepatic lipid disturbances derived from a high fat diet’ reported the effects of Chenopodium quinoa’s ingredients on major lipids hepatic profile and functional selective differentiation of monocyte-derived macrophages and innate lymphoid cells in mice under a high fat diet. This work in interesting. I have no specific concern, but it is mandatory to correct the manuscript in this point:

Please unify whether the Latin name is italicized. ‘C. quinoa’ in the whole text should be italicized.

Many thanks for your comment. According to the reviewer’s comment, the name ‘C. quinoa’ has been italicized throughout the manuscript.

Reviewer 3 Report

The article discusses the effects of Chenopodium quinoa's ingredients on hepatic lipid profile and immune cells in mice fed a high-fat diet. The study found that the low-molecular-weight protein fraction (LWPF) and lipid fraction (qLF) from quinoa improved hepatic lipid utilization and reduced hepatic lipid accumulation. These ingredients also influenced hepatic innate immunity and gut microbiota composition. The study suggests that quinoa's ingredients have immunonutritional properties that can benefit immune and metabolic imbalances caused by a high-fat diet.

Major concerns:

1.      Could the authors explain why they used Rag2-/- and also Rag2-/- IL2-/-. I did not see any article mentioned Rag2-/- IL2-/-. Is there any specific feature of this kind of mice?

2.      Please explain the different results between these two mice, especially to ILC2 and Macrophage. And try to find out the mechanism.

3.      Gating strategy and representative figures for ILC2 and Macrophage should be present

4.      All the data should be present instead of the option data not shown

5.      Missing the function of ILC2 and macrophage. Phenotype is not sufficient

Author Response

Reviewer #3

Comments and Suggestions for Authors

The article discusses the effects of Chenopodium quinoa's ingredients on hepatic lipid profile and immune cells in mice fed a high-fat diet. The study found that the low-molecular-weight protein fraction (LWPF) and lipid fraction (qLF) from quinoa improved hepatic lipid utilization and reduced hepatic lipid accumulation. These ingredients also influenced hepatic innate immunity and gut microbiota composition. The study suggests that quinoa's ingredients have immunonutritional properties that can benefit immune and metabolic imbalances caused by a high-fat diet.

Major concerns:

  1. Could the authors explain why they used Rag2-/- and also Rag2-/- IL2-/-. I did not see any article mentioned Rag2-/- IL2-/-. Is there any specific feature of this kind of mice?

In page 2, line 67 it is stated ‘Besides, intestinal innate lymphoid cells (ILCs) have been revealed as key contributors to diet-induced obesity [4].’. All info concerning the preclinical models, which is out of the scope of this journal can be found in this reference. These authors used both models to define the role of ILCs, lacking in the Rag2-/-IL2-/- mice but not in Rag2-/- mice as it is also indicated in the abstract. The lack of ILCs in Rag2-/-Il2-/- mice makes them resistant to diet-induced obesity due to the determinant role of ILCs enabling the absorption of dietary fat.

  1. Please explain the different results between these two mice, especially to ILC2 and Macrophage. And try to find out the mechanism.

Many thanks for your comment, which we really appreciate. The reviewer raised up a critical question concerning the role of lipids to influence, worsening or improving, the inflammatory processes and thereby, the activation of proinflammatory macrophages. These processes would affect their role to modulate the hepatic fat accumulation (References 5 and 10 in the manuscript).

A major aspect of our research line tries to identify the underlying molecular mechanisms to explain the differential aspects derived from the different C. quinoa’s ingredients. In this line, it is stated in the manuscript (page 12, line 421-425) ‘From a molecular perspective it could be hypothesized that the LMWF due to its capacity to interact with TLR4 [11] seems to influence, among other, either the release or activity of the soluble epoxide hydrolase in peripheral blood [26].’ The implication of this receptor could link the fat/lipid absorption to the macrophage activation. However, currently, we have not fully defined this mechanism. This effect seems to be significantly relevant for ILC2s, which in fact are critical for the NAFLD development; ‘…ILCs have been identified as critical determinants in the diet-induced obesity [4]. Particularly, a major role was attributed to intestinal ILC2s…’ (page 11, line 367-368). Currently, we only can suggest these connections and this is the reason why it was decided to state the info within the manuscript but to speculate too much.

  1. Gating strategy and representative figures for ILC2 and Macrophage should be present

We fully appreciate your comment. The reason why this strategy hasn’t been indicated is because this is out of the scope of the journal. All information about the strategy to identify macrophages can be already found in reference 16 as well as that for ILCs in ref 15. Otherwise, we have to apologize because the reference to identify ILCs is already missing in the manuscript.

In the light of the reviewer’s comment a new reference describing the phenotype of ILCs has been introduced in the manuscript; Bal SM, Golebski K, Spits H. Plasticity of innate lymphoid cell subsets. Nat Rev Immunol 2020;20:552–65. doi:10.1038/s41577-020-0282-9

  1. All the data should be present instead of the option data not shown

Many thanks for your comment. The reason to not include the ‘maximal respiration’ was the number of panels already included in the Figure 4.

In view of the reviewer’s comment a new panel has been included showing the effects of the ‘maximal respiration’ of macrophage (Page 8, line 274).

  1. Missing the function of ILC2 and macrophage. Phenotype is not sufficient.

We feel sorry for this inconvenience. As already stated in the introduction (references 4 and 5), ‘…the role of ILC2s still remains imprecise and appear to affect, either worsening or improving, NAFLD/obesity [21].’ ILC2s activation causing negative effects in NAFLD development would occur via the profibrotic IL-13 as well as the take up of lipids (page 12, line 368-378). However, ‘…only Rag2-/- mice fed with the LMWF showed a sig-nificantly decreased hepatic TGs fraction (Fig. 2F) and total lipids content (Fig. 3A) associated to a significant expanded ILC2s proportion (Fig. 4F). Besides, the increased CD117+Nkp46+CD56+ cell population supports the accumulation and development of a M1-like (CD68+F4/80+) phenotype by hepatic macrophages [23]’. Altogether, data transpire in the sense that C. quinoa’s LMWF influences the macrophage functional differentiation that prevents ILC2s activation towards negative effects, while favoring their function controlling fat accumulation.

We fully understand the reviewer’s position, and we fully agree with it. Unfortunately, this study only approaches the potential of C. quinoa’s ingredients to modulate the hepatic immunity. In order to identify the specific function of ILC2s it would be needed to transfer such population to the Rag2-/-Il2-/- model, lacking of those and only producing inmature precursors. Besides, to identify the specific function of macrophages it would be needed to administer the animals with liposomes based on clodronate to eliminate the resident macrophages and confirm that only those develop under the treatment with C. quinoa are responsible for the outcomes.

Round 2

Reviewer 3 Report

Although the author provided many explanations, they did not adequately address my question.

Firstly, the author used Rag2-/-IL2-/- mice, but I have not seen this type of mouse mentioned in any literature. If the author obtained this mouse from another source and is reporting it for the first time in the paper, a detailed strategy for the creation of the mouse and its phenotype should be provided. Otherwise, this doesn't appear to be a rigorous experiment. If this mouse type has been reported before, please cite the relevant literature so that everyone can be aware of its characteristics. Furthermore, I'm unclear on the reason ILC2 cells are not present in the Rag2-/-IL2-/- mice. The author must provide an explanation.

Secondly, I always believe that the gating strategy and representative images are of utmost importance. The author cited literature to validate their gating strategy, one of which is a review article. So, how exactly did the author gate the target cells? Are there representative images? These are essential components of a paper and they don't conflict with the scope of the journal.

The author is urged to seriously address the two aforementioned comments.

Author Response

Although the author provided many explanations, they did not adequately address my question.

 Firstly, the author used Rag2-/-IL2-/- mice, but I have not seen this type of mouse mentioned in any literature. If the author obtained this mouse from another source and is reporting it for the first time in the paper, a detailed strategy for the creation of the mouse and its phenotype should be provided. Otherwise, this doesn't appear to be a rigorous experiment. If this mouse type has been reported before, please cite the relevant literature so that everyone can be aware of its characteristics. Furthermore, I'm unclear on the reason ILC2 cells are not present in the Rag2-/-IL2-/- mice. The author must provide an explanation.

We feel sorry for this misunderstanding. The reviewer can read about Rag2-/-IL2-/- mice in the literature quoted in the manuscript (Sasaki, T.; Moro, K.; Kubota, T.; Kubota, N.; Kato, T.; Ohno, H.; Nakae, S.; Saito, H.; Koyasu, S. Innate Lymphoid Cells in the Induction of Obesity. Cell Rep. 2019, 28, 202-217). This reference was already quoted in the manuscript from the first version of the manuscript. It is worth to bring here that we have not developed these mice strains; we just took those from the literature.

Rag2-/-IL2-/- mice were introduced by Koyasu and Moro,2012 (Front Immunol. 2012; 3: 101; doi: 10.3389/fimmu.2012.00101) and Moro and Koyasu,2015 (Semin Immunopathol 2015 Jan;37(1):27-37. doi: 10.1007/s00281-014-0470-4). These authors indicate the deficiency of ILCs, all subphenotypes, but not only ILC2s.

The review already provided in the manuscript (15. Bal, S.M.; Golebski, K.; Spits, H. Plasticity of innate lymphoid cell subsets. Nat Rev Immunol 2020, 20, 552.), which we forgot in the first version, shows the development of ILCs subsets from common precursors; all ILCs come from a CD117+ inmature ILC.

Secondly, I always believe that the gating strategy and representative images are of utmost importance. The author cited literature to validate their gating strategy, one of which is a review article. So, how exactly did the author gate the target cells? Are there representative images? These are essential components of a paper and they don't conflict with the scope of the journal.

As stated in the previous comment, the review provided shows the development of ILCs subsets (Bal, S.M.; Golebski, K.; Spits, H. Plasticity of innate lymphoid cell subsets. Nat Rev Immunol 2020, 20, 552) and states the markers for the different ILC subsets. Thus, taking into consideration the development process the rationale for the gating strategy has been, 1st identify CD117+ cells using a bi-parametric plot together with CD56+ cells, and 2nd the CD117+CD56- population was further analyzed for CD117+KLRG1+.

Similarly, to identify the macrophage population it has been 1st identify the CD11b population together with F4/80+ in a bi-parametric plot, the CD11b+F4/80+ population was further analyzed for CD68+ and TLR4+ in another bi-parametric plot.

The reason why we have used these combinations has been the nature of the fluorochromes carried by the antibodies used, which are commercially available. In case that the reviewer decides to order ‘customized’ antibodies carrying different fluorochromes, the strategy would be different. This is an additional reason why we did not initially include these panels.

In view of the reviewer’s comment. Two new panels have been introduced in Figure 4 showing typical plots for the CD117+ vs KLRG1+ population and CD68+ vs F4/80+ cells.

The author is urged to seriously address the two aforementioned comments.

Many thanks for your comments, which we really appreciate.